# Development of microfluidic chip for entrapping tobacco BY-2 cells

**Kazunori Shimizu**[1,2]☯*, **Yaichi Kawakatsu**[3]☯, **Ken-ichi Kurotani**[3]☯, **Masahiro Kikkawa**[1], **Ryo Tabata**[2,4], **Daisuke Kurihara**[5,6], **Hiroyuki Honda**[1], **Michitaka Notaguchi**[2,3,4,6]*

**1** Department of Biomolecular Engineering, Graduate School of Engineering, Nagoya University, Nagoya, Japan, **2** Institute for Advanced Research, Nagoya University, Nagoya, Japan, **3** Bioscience and Biotechnology Center, Nagoya University, Nagoya, Japan, **4** Graduate School of Bioagricultural Sciences, Nagoya University, Nagoya, Japan, **5** JST PRESTO, Nagoya University, Nagoya, Japan, **6** Institute of Transformative Bio-Molecules (ITbM), Nagoya University, Nagoya, Japan

☯ These authors contributed equally to this work.
* shimizu@chembio.nagoya-u.ac.jp (KS); notaguchi.michitaka@b.mbox.nagoya-u.ac.jp (MN)

**Data Availability Statement:** All relevant data are within the paper and its Supporting Information files.

**Funding:** This work was supported by grants from the Japan Society for the Promotion of Science Grants-in-Aid for Scientific Research (18K05373

## Abstract

The tobacco BY-2 cell line has been used widely as a model system in plant cell biology. BY-2 cells are nearly transparent, which facilitates cell imaging using fluorescent markers. As cultured cells are drifted in the medium, therefore, it was difficult to observe them for a long period. Hence, we developed a microfluidic device that traps BY-2 cells and fixes their positions to allow monitoring the physiological activity of cells. The device contains 112 trap zones, with parallel slots connected in series at three levels in the flow channel. BY-2 cells were cultured for 7 days and filtered using a sieve and a cell strainer before use to isolate short cell filaments consisting of only a few cells. The isolated cells were introduced into the flow channel, resulting in entrapment of cell filaments at 25 out of 112 trap zones (22.3%). The cell numbers increased through cell division from 1 to 4 days after trapping with a peak of mitotic index on day 2. Recovery experiments of fluorescent proteins after photobleaching confirmed cell survival and permeability of plasmodesmata. Thus, this microfluidic device and one-dimensional plant cell samples allowed us to observe cell activity in real time under controllable conditions.

## Introduction

Plants that flourish on Earth are multicellular organisms and have a system that differentiates into various organs. In multicellular organisms, cells share substances necessary to sustain life and share information to function as an organism. Plants exchange materials and information between individual cells, and also transport materials systemically between distant organs throughout the body. In higher plants, the transport of substances and signals over long distances is via vascular bundles, which are specially differentiated pathways in xylem and phloem [1, 2]. These long-distance transports also rely on cell-to-cell transport mechanisms for loading of substances into the vascular bundle and for unloading at the destination [3]. Thus, cellular events are the basis both for local and systemic transport and signaling.

The cultured cells are expected to be useful in the analyses of plant cell signaling and material transfer because the cells are arranged in one-dimensional position, allowing for clear

and 20H05501 to RT, 18KT0040, 18H03950, 21H00368, and 21H05657 to MN and 20H03273 to KK and MN), Grant-in-Aid for Scientific Research on Innovative Areas (20H05358 to DK), the Japan Science and Technology Agency PRESTO program (JPMJPR18K4 to DK), and the Canon Foundation (R17-0070 to MN). The funders had no role in study design, data collection and analysis, decision to publish, or preparation of the manuscript.

**Competing interests:** The authors declare that the research was conducted in the absence of any commercial or financial relationships that could be construed as a potential conflict of interest.

observation. BY-2 cells are cultured plant cells derived from *Nicotiana tabacum* L. 'Bright Yellow 2' [4]. Because of their high proliferative potential, which doubles in 12–14 hours, and the facilitative nature of transformation using Agrobacterium, they are widely used as a model system in plant cellular and physiological studies including for cell cycle, cytokinesis [5, 6]. BY-2 cells can be cultured in large numbers like a discrete unicellular organism, but they are also noted to divide unidirectionally at an angle of 90 degrees to longitudinal axis of the cell, and under suitable culture conditions, a dozen or more cells can be cultured in a vertically connected state while maintaining the cell lineage of division. These connected cells enable us to study the movement of proteins and other substances between cells and the transduction of signaling molecules such as plant hormones.

Plants have evolved an intracellular communication pathway through microscopic tunnels that penetrate the cell wall and directly connect the cytoplasm of neighboring cells. These tunnels, called plasmodesmata (PDs), are ultrafine structures, often as small as 30 nm in diameter and 100 nm length, and facilitate the movement of materials between cells [7]. They are lined by the plasma membrane (PM) with a tubular endoplasmic reticulum (ER) inside and the space between the PM-ER is thought to control plasmodesmata permeability [8]. It is estimated that about 1,000–100,000 PDs are located between neighboring cells [9–11]. Recent studies have shown that PDs are involved in the transport of proteins, mRNAs, plant hormones, and other substances involved in plant functions. They also allow the transfer of pathogenic bacteria, viruses and fungi between cells [7, 12–16]. Intercellular localization of PD localized proteins has been studied using cultured cells [17]. The PD-enriched wall fraction from the BY-2 cell line leads to isolation of PD-related factors such as a Non-Cell-Autonomous-Protein-Pathway1 (NCAPP1) [18], a Plasmodesmata Germin-like protein (PDGLP1) [19] and a PD-Associated Protein Kinase (PARK) [20]. They have been also used to investigate intercellular localization of viral proteins such as viral movement proteins [21]. Thus, this model system has been used to identify the molecules associated with PDs.

In this study, we aimed to establish a method to assess the plant cell activity and cell-to-cell molecular transport using a microfluidic device in which cultured plant cells can be fixed in position. Since BY-2 cells can be transformed with high efficiency via Agrobacterium, it is possible to monitor cell-to-cell movement of proteins by expressing proteins fused with fluorescent proteins such as GFP. Culturing BY-2 cells in the dark suppresses the development of plastids in cells, resulting in a colorless state that allows for highly sensitive detection. On the other hand, since the cells are not adherent and the liquid culture medium needs to be constantly stirred to supply nutrients and oxygen, BY-2 cells are suspended in the culture medium, making it difficult to observe the same cell filaments for an extended time period. Moreover, fixing the position of the object is important especially for high-resolution imaging. First, we observed the properties of BY-2 cells, including their length and linearity, during culture to select the appropriate duration of culture for cells used in this study. Next, to avoid changing the position of BY-2 cells during the experiments, we fabricated a microfluidic device with a microchannel to trap the BY-2 cells. The viability and mitotic activity of BY-2 cells entrapped in this device were examined by measuring the mitotic index. Finally, we explored the permeability of PDs in trapped BY-2 cells by quantitatively assessing fluorescence recovery after photobleaching (FRAP) to evaluate the usability of the device for studies of cell-to-cell communication in plant cells.

## Materials and methods

### Culturing of tobacco BY-2 cells

Tobacco BY-2 cells were cultured in modified Linsmaier and Skoog (LS) medium (see the following recipe) [22]. The BY-2 cell cultures were maintained at 26˚C with shaking at 130 rpm

in the dark on a shaking stirrer (NR-3, TAITEC CORPORATION, Saitama, Japan) in an incubator (IS-2000, Toyo Seisakusho Co., Ltd., Chiba, Japan). The medium was dispensed into 300-mL flasks (95 mL medium/flask). Each flask was capped with a silicon stopper, and autoclaved at 120°C for 20 min. For subculture, 3 mL cell suspension was removed from the flask on the 7th day of culture and transferred to a new flask. For incubation of the GFP and 3×Venus-NLS strains, 50 mg/L kanamycin sulfate (117–00341, Fujifilm Wako Pure Chemicals Co., Ltd., Osaka, Japan) was added to the culture medium.

## Recipe for modified LS medium

- Murashige and Skoog plant salt mixture (392–00591, Fujifilm Wako), 4.6 g/L

- Sucrose (193–00025, Fujifilm Wako), 30 g/L

- Potassium Dihydrogen Phosphate (169–04245, Fujifilm Wako), 0.2 g/L

- Myo-inositol (I5125, Sigma, St. Louis, MO, USA), 0.1 g/L

- Thiamin hydrochloride (205–00855, Fujifilm Wako), 1.0 mg/L

- 2.4-dichlorophenoxyacetic acid (040–18532, Fujifilm Wako), 0.2 mg/L

Adjust pH to 5.8 with KOH

## Plasmid construction and transformation of tobacco BY-2 cells

For the *35S::EGFP* construct (designated as MNv307), *EGFP* fragment was cloned into the pENTR/D-TOPO vector (Thermo Fisher Scientific, Waltham, MA, USA) and introduced into the binary vector pGWB2 [23]. For the *GmHSP:3×Venus-NLS* construct (designated as MNv216), *GmHSP*/pGWB2 binary vector was firstly generated. The 446-bp upstream region (−506 to −61 bp) of soybean *Gmhsp17.3-B* [24] was amplified by PCR with primers, 5′-CAC CaagcttTAGTCAGCCTTTTAAGAGATAG-3′ and 5′-tctagaCCggtaccGTCGACCTA CAAAACTGCTAAC-3′, and cloned into the pENTR/D-TOPO vector. The fragment was digested by *HindIII* and *XbaI* and introduced into the same sites of pGWB2 to generate *GmHSP*/pGWB2. Secondary, the *3×Venus-NLS* fragment was cloned into the pENTR/ D-TOPO vector and introduced into the binary vector *AtHSP*/pGWB2 by an LR reaction. To generate transgenic BY-2 cell lines expressing either GFP or 3×Venus-NLS, *Agrobacterium*-mediated transformation was performed as previously described [25]. Transformants were selected on modified LS medium containing 1.5% (w/v) agar and 50 μg/mL kanamycin and then cultured for 3 weeks before initiating a liquid culture in modified LS medium. The callus cells were transferred to liquid medium using a sterilized platinum loop. The cells were subcultured once or twice before use to stabilize growth. The *GmHSP::3×Venus-NLS* transgenic cell line constitutively express fluorescent proteins without heat shock treatment in medium culture.

## Microdevice fabrication

The microfluidic devices were fabricated by replica molding using polydimethylsiloxane (PDMS; SILPOT 184, DuPont Toray Specialty Materials K.K., Tokyo, Japan) as we reported previously, with some modifications [26]. Briefly, the silicon wafer was spin-coated with SU-8 3050 (MicroChem Corp., Newton, MA, USA) at 1750 rpm for 60 s and baked at 95°C for 30 min. The wafer was exposed to ultraviolet light through a photomask and baked at 95°C for 10 min. This operation resulted in the formation of grooves with a depth of 50 μm. After

development using SU-8 developer (MicroChem Corp.), PDMS mixed with a curing reagent at a ratio of 10:1 was poured onto the mold to make it about 2 mm thick, and then cured by baking at 70˚C for 60 min. Two 1.5-mm diameter holes were made for an inlet and outlet using a biopsy needle (Kai Industry, Gifu, Japan). Then, the PDMS and a slide glass (S021230, MATSUNAMI, Osaka, Japan) or a cover glass (Thickness NO.1, MATSUNAMI) were treated with air plasma (SEDE-P, Meiwafosis, Osaka, Japan) and bonded to each other. For FRAP experiments, cover glasses were used.

## Preparation and image analysis of BY-2 cells

To prepare short filaments of BY-2 cells, the BY-2 cell culture was passed through a 140-μm sieve (NRK, Tokyo, Japan) and subsequently through 70 and 40-μm cell strainers (pluriSelect Life Science, Leipzig, Germany). Before and after separation, a 1-mL aliquot of BY-2 cell culture was mixed with CellTracker Green CMFDA Dye (C7025, Thermo Fisher Scientific, Tokyo, Japan) at 5 μM and incubated for 30 min at 26˚C with agitation. Then, 10 μL BY-2 cell-containing medium was placed on the slide glass and covered with a cover glass. Fluorescence images were obtained using a fluorescence microscope (BZ-X700, Keyence, Osaka, Japan). The images were analyzed by ImageJ; the objects were recognized by color threshold and subsequently the objects crossing the edges were excluded. Then, the minimum length of the minor axis of the approximate minimum ellipse and area of the objects, *i.e*., BY-2 cell filaments, were measured. These procedures are illustrated in S1 Fig.

## Introducing BY-2 cells into the microfluidic device

The device was sterilized by exposure to ultraviolet light for 1 h and treated with air plasma to make the surface of the microchannels hydrophilic. Then, an empty filter tip was stuck to the outlet, and 700 μL medium without cells was slowly introduced into the inlet and allowed to fill the channels. If any air remained in the device, the device was left for about 30 min until the air disappeared. The cell suspension was prepared at a concentration of 500–1000 cell filaments/mL. A 700-μL aliquot of the cell suspension was introduced slowly into the channels from the inlet. Then, a 700-μL aliquot of medium without cells was introduced to remove the un-trapped cells.

## Culture of trapped BY-2 cells in the microfluidic device

After introducing cells into the channels, the inlet and outlet of the microchannels were plugged with P200 filter pipette tips (127-200XS, Watson, Tokyo, Japan) filled with/without culture medium to prevent the channels from drying. Then, the device with tips was incubated at 26˚C in the dark. The medium was exchanged every 24 h by introducing 700 μL medium slowly from the inlet.

## Mitotic index measurement

Three culture conditions of BY-2 cells were prepared to measure mitotic indexes; (1) a conventional shake culture, (2) a shake culture of the short filaments, and (3) a culture of the short filaments in the microfluidic devices. For the conventional shake culture, 3 mL of 7 days cell culture suspension was transplanted to a new flask containing 100 mL of culture media and cultured. For the latter two conditions, short filaments of BY-2 cells were sorted from 7 days culture using cell strainers as described above and conditioned at a concentration of 500–1000 cell filaments/mL. For the shake culture of the short cell filaments, 100 mL of the cell suspension was cultured in the same manner with the conventional shake culture. For the culture of

the short filaments in the devices, the short filaments were introduced to the microchannels as described above and cultured until analysis. For samples of the shake cultures (1) and (2), 10 mL of cell suspension was collected each day to analyze. The short filaments were collected by centrifuge at 400 × g for 15 min. For mitotic index measurement, the cells were fixed with 1% glutaraldehyde for 30 min and stained nuclei with 2 µg/mL of 4',6-diamidino-2-phenylindole (DAPI). Loading of the solutions into the microchannels was performed in the same manner with that for cell suspension. The number of cells undergoing mitosis was counted (n = 337–574 for each time point).

### Assessment of PD permeability

A culture of BY-2 cells was prepared at 500–1000 cells/mL as described above. To avoid unintended physical stress during the cell trapping procedure that may close PDs, and to increase the number of filaments with three or more cells, we cultured the trapped BY-2 cells for 1 day or longer.

A confocal microscope (LSM510/LSM5 Pascal, Axiovert, Zeiss, Germany) equipped with 488 nm argon laser (LGK7872ML, LASOS Lasertechnik GmbH, Jena, Germany) was used for observation and photobleaching. For photobleaching, the settings were as follows: objective lens, Plan-Neofluor 10x/0.3; pinhole, max (141.4 µm); laser power, 100%; observation area, 898.2 µm × 898.2 µm; laser irradiation area, 219.1 µm × 219.1 µm; photobleaching period by laser scanning, 963.04 msec; irradiation duration, 20 min for 3xVenus-NLS and 20–35 min for GFP, depend on the expression levels. We obtained fluorescence images of the trapped targeted filament three to seven times (at least, before, just after, and 200 min, and additionally 20, 40, 60, 120, 140, 160, 180 min and 24 hours after photobleaching) with a 505 nm long pass filter. LSM consecutive optical sections including entire nucleus were obtained and merged. Image analysis was performed using ImageJ. To calculate the amount of recovery of the fluorescence signal after photobleaching, the fluorescent intensity of each time point was divided by the value before photobleaching.

## Results and discussion

### Analysis of shapes of BY-2 cells during culture

A model cell biology system in plants, BY-2 cell line forms short cell filaments consisting of several cylindrical cells connected in tandem in liquid culture medium (Fig 1A). Established cell culture protocols, high efficient transformation techniques, and transparency of the BY-2 cells enable cell imaging and detailed analysis. Thus, we anticipated that a trapping method of BY-2 individual cells will enhance traceability of cell physiology and activity through a long period observation. To design a microfluidic device that can fix the position of BY-2 cells, we observed the shapes of BY-2 cells. The BY-2 cell filaments were linear but not uniform in shape and size. In particular, the degree of bending was different in each filament (Fig 1B). Since BY-2 cells cultured in flask batches were passaged every 7 days, we counted the number of cells in one filament on days 3, 5, and 7. The percentages of filaments containing 10 or more cells were 69.2% (27/39) on day 3, 58.8% (20/34) on day 5, and only 14.0% (8/57) on day 7. The percentage of filaments containing four or fewer cells increased to 64.9% (37/57) on day 7 in our conditions (Fig 1C). The decrease in the percentage of filaments with 10 or more cells on day 5 and day 7 was significant at $p<0.01$ by Fisher's exact test. Since the mitotic index usually peaks at around days 2 and 3 and declines rapidly [27], the prolonged cell culture period might have caused that long cell filaments break down into the short filaments.

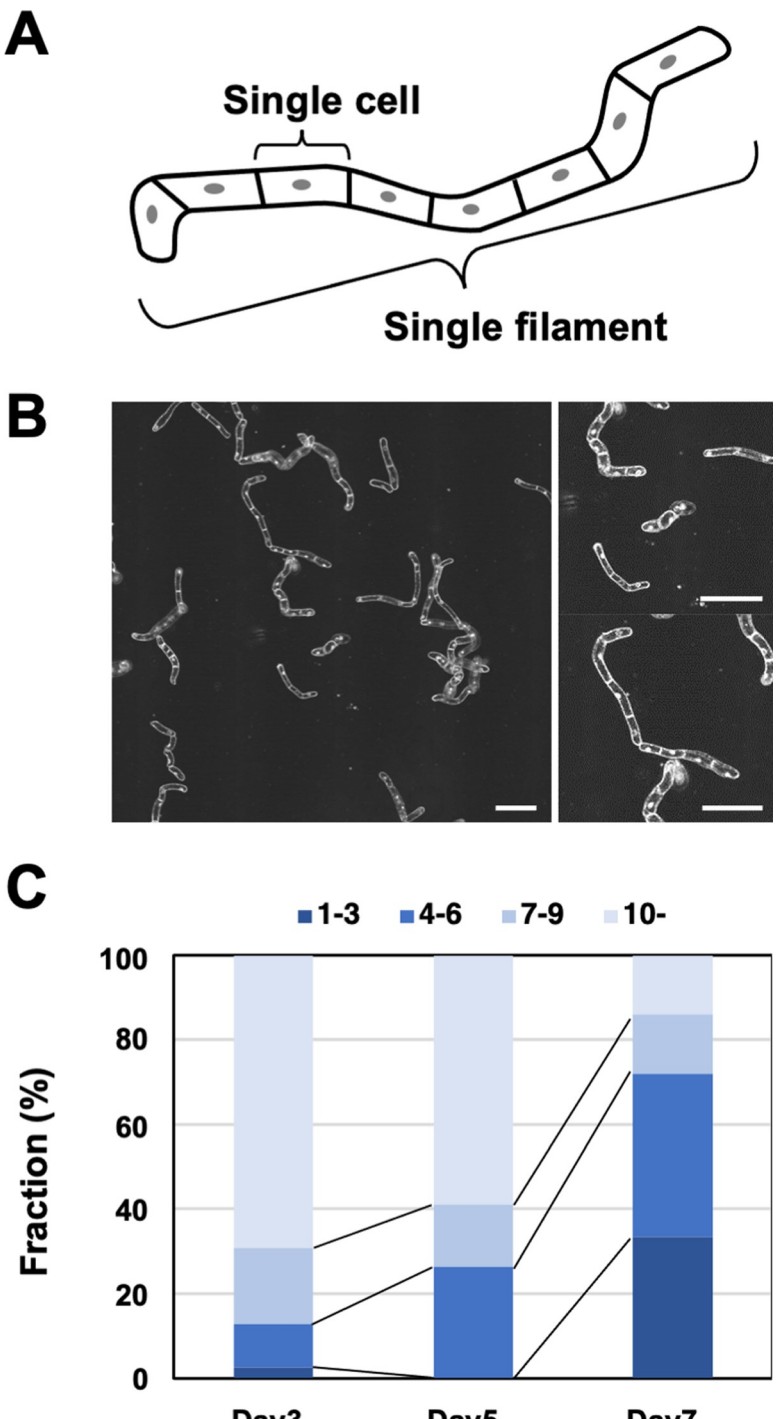

**Fig 1. Properties and shapes of BY-2 cells.** (A) Schematic image of a BY-2 cell. (B) Representative images of BY-2 cells under phase-contrast microscope. Scale bars, 200 μm. (C) Distribution of number of cells in a single filament of BY-2 cells on days 3, 5, and 7 of culture. Total number of filaments counted at day 3, 5 and 7 were 39, 34 and 57, respectively.

## Design and fabrication of microfluidic device for fixing the position of BY-2 cells

We designed and fabricated a microfluidic device for fixing the position of BY-2 cells referring to previous report [28] (Fig 2A). In the previous report, cell sorting and trapping were demonstrated using polystyrene microspheres (sizes: 15 μm, 6 μm, and 4 μm) and three different waterborne parasites including *Giardia* cysts (ellipsoid with short and long axes of 8 μm and 19 μm, respectively). To trap BY-2 cells, in this study, we modified the dimensions and the shape of the trap zones in the microfluidic device. The device has a main channel and a side channel that are connected by a trap zone (Fig 2B). The cell suspension was added to the main channel through the inlet. According to the simulation by Kim and colleagues, the fluidic pressure of the main channel is always higher than that of the side channel at the same position because the ratio of the width of the inlet and outlet of the trap zone, the main channel, and the side channel is set to 2:1:8:20 [28].

In this study, we designed each trap zone to entrap one short filament of BY-2 cells (Fig 1C). The entrance of the trap zone was 50 μm, which is almost the same width of a single filament of BY-2 cells. The exit of the trap zone was 25 μm, which is narrower than the cell width. The length of the trap zone was set to 300 μm, which is about the length of a filament of several cells. The device had a total of 112 trap zones (Fig 2B).

## Introducing BY-2 cells into the microfluidic device

We examined whether short filaments of BY-2 cells could be trapped in the developed device. To clearly observe the location of cells in the device, the cells were stained by CellTracker Green before use. We introduced an aliquot of cells cultured for 7 days, which contained many short cell filaments, into the inlet (Fig 1C). However, the cells became clogged in the main channel near the inlet and were not trapped in the trap zone. Microscopic observations revealed that this was caused by clogging of the main channel with long and bent cell filaments in the cell suspension. To remove the long and bent filaments, the cell culture was passed through a sieve and a series of cell strainers (see Materials and Methods). The cell suspension after this separation step contained single cells and shorter and straighter cell filaments (Fig 3A and S2 Fig). These single cells or filaments consisted of 1–4 cells and the percentage of those was around 30% in the original cell culture at day 7 (Fig 1C). When the separated cells were introduced into the device, the cells in short filaments were trapped in the trap zone (Fig 3B). A 700-μL aliquot of the cell suspension at a concentration of 500–1000 cell filaments/mL was introduced slowly into the channels from the inlet. Because further cell loading into the channels by concentrating cell suspension or increasing loading volume arose cell clogging of the channels, we used this condition. On average, 25 ± 4.5 cell filaments were trapped by the device with 112 trap zones (entrapment frequency, 22.3% ± 4.1%). In a rough calculation, around 5% of the separated cell filaments were trapped and 95% of the separated cell filaments remain trapped. The number of cells in the trapped filament at the trap zone was counted. More than half of the filaments consisted of two cells, the other filaments consisted of one cell or three cells or more (Fig 3C). Thus, the short filaments were successfully entrapped by the microfluidic devices. The width of the trap zone was determined from the standard short axis size of BY2 cells in the 1-cell state. The efficiency to trap BY-2 cell filaments in the trap zones was smaller than previously reported for *Giardia* cysts, probably because of the geometry complexity of filament-forming BY-2 cells [28].

## Cultivation of trapped BY-2 in the microfluidic device

To examine cell physiology and behaviors using the trapped BY-2 cells, it was important that the cells remained undamaged. We therefore monitored the trapped BY-2 cells over several

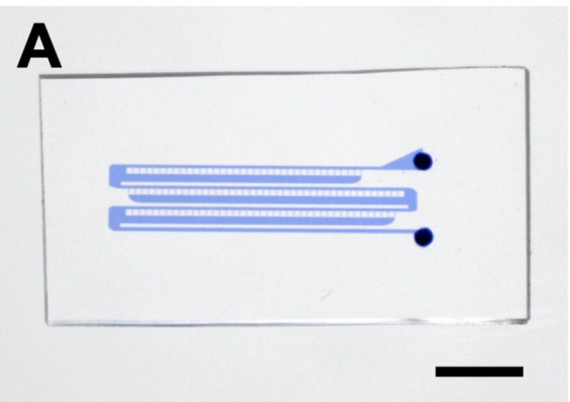

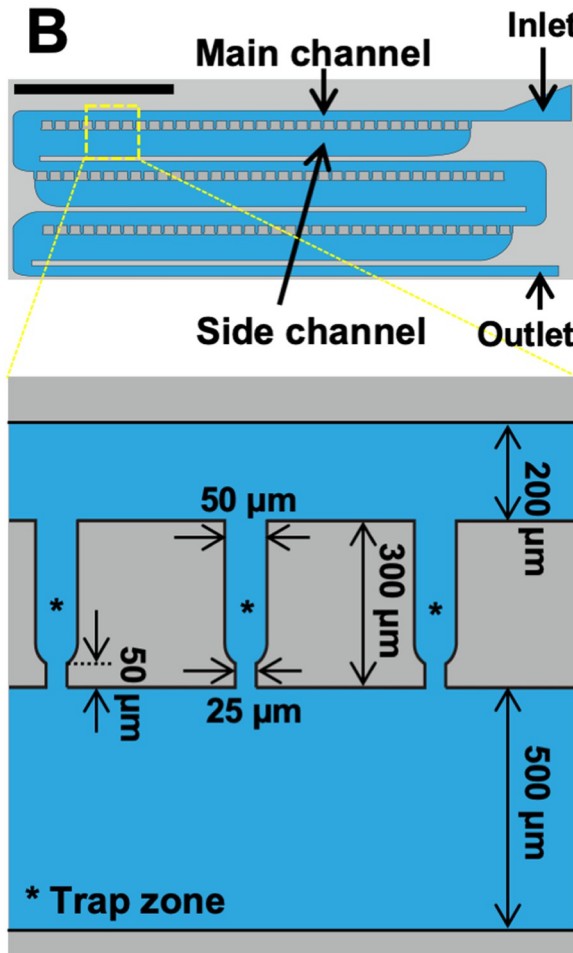

**Fig 2. Design and fabrication of microfluidic device.** (A) Fabricated microdevice with channels filled with blue-colored water. Scale bar, 5 mm. (B) Shape and size of microchannels.

days to see whether they retained cell proliferation ability in the device. As shown in Fig 4A and 4B, the number of cells in the trapped single filament at the trap zone increased over time through cell division and the filament became longer. Observation of BY-2 cells expressing

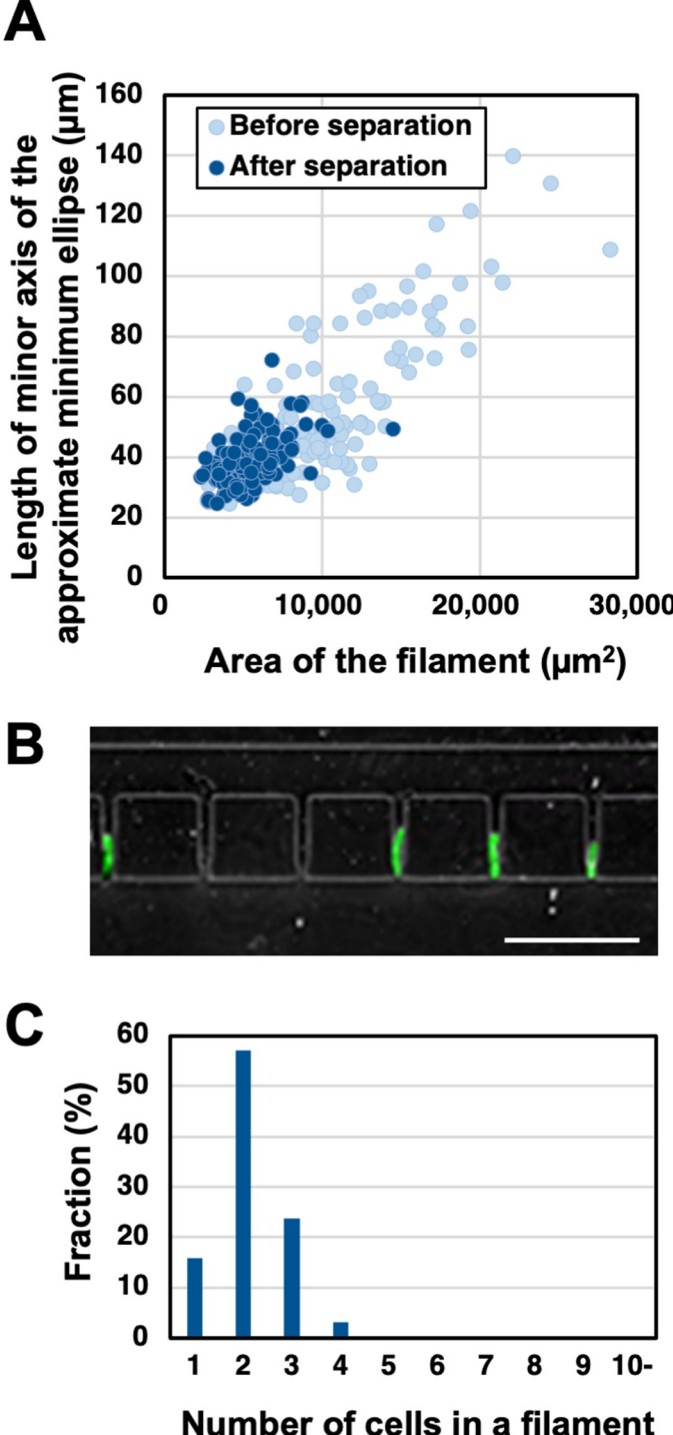

**Fig 3. Trapping of BY-2 cells in the microfluidic device.** (A) Relationship between length of minor axis of approximate minimum ellipse and area of cell filament. Light blue circle: before cell separation. Dark blue-filled circle: after cell separation. BY-2 cells stained with CellTracker Green were used and 150 filaments were analyzed for each condition. (B) Representative image of trapped cells. Scale bar, 500 μm. (C) Fraction of number of cells in the trapped filaments in microfluidic devices.

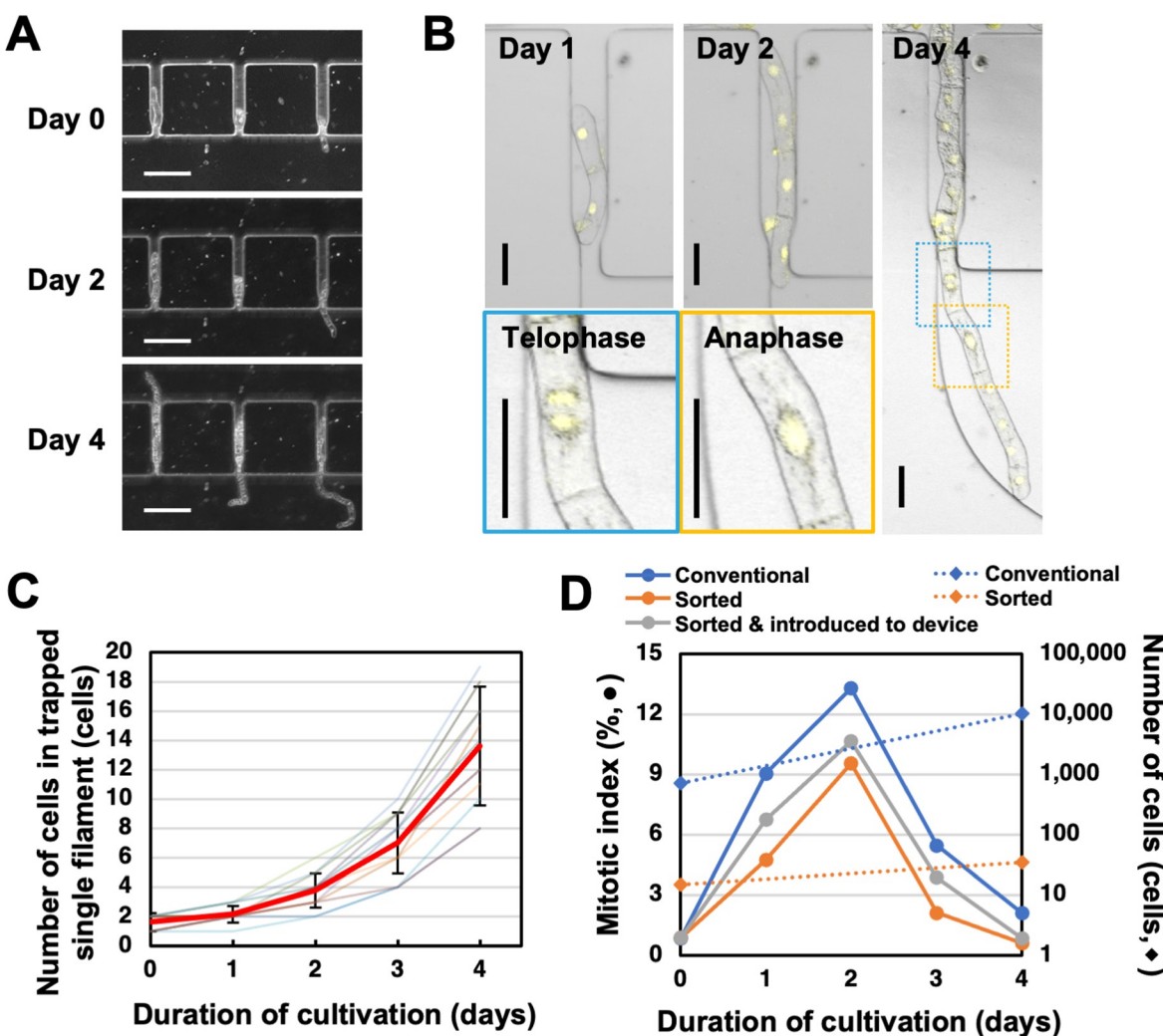

**Fig 4. Culturing of trapped BY-2 cells inside the microfluidic device.** (A) Representative image of trapped cells on days 0, 2 and 4 after trapping. Scale bars, 200 μm. (B) Laser scanning microscope image of BY-2 cells expressing 3xVenus-NLS in the microfluidic device on days 1, 2 and 4 after trapping. Cells in mitosis are surrounded by dashed squares in Day 4, and magnified views of these regions are shown. The same color represents the same area. Confocal laser scanning microscopy (LSM) images were merged from 8 consecutive optical sections. Fluorescence images were merged with bright-field images. Scale bars, 50 μm. (C) Change in number of cells in a trapped single filament. The cell number of 13 filaments were counted (Light-colored lines). The red line represents the average number of cells in 13 filaments. Error bars indicate SD. (D) The mitotic indexes of conventional culture conditions, cells after sorting, and sorted cells cultured inside the microfluidic device (circles and solid lines) (n = 337–574 for each time point). For the previous two, the counts of cells on day 0 and 4 are plotted together (diamonds and dashed lines).

nuclear localized 3×Venus-NLS showed that cell divisions occurred both in cells located at the end of the filament and in cells at inside position of the filament (Fig 4B and S3 Fig). We rarely observed vertically formed daughter cells. The average number of cells in one trapped filament was 1.6 ± 0.5 immediately after trapping and increased to 13.6 ± 3.8 after 4 days of culture (Fig 4C). We measured mitotic indexes of a conventional shake cell culture, a shake cell culture of the sorted short filaments and the short filaments introduced into the devices from day 0 to day 4. The mitotic indexes of the short filaments in shake culture and in the microchannels were slightly less than the conventional culture but both showed similar patterns overtime with the conventional culture which peaked at day 2 (n = 337–574, Fig 4D). These results confirmed that the trapped BY-2 cells in this device can proceed cell proliferation.

We calculated the percentage of cell filaments that remained trapped until day 4 (number of filaments trapped at the trap zone on day 0 / number of filaments trapped at the trap zone on day 4 × 100). This analysis revealed that 61.6% ± 7.2% of cell filaments remained trapped on day 4. It is likely that the medium flow released some of the trapped cell filaments from the trap zone during 4 days of culture. Although it was not determined how the trapped cell filaments were actually lost, a large and fast flow of liquid medium during pipetting may cause removal of the trapped BY-2 cell filaments from the entrapped zones. In future studies, it should be possible to adjust the trap zone shape and precisely control the flow velocity of the medium to increase the percentage of trapped cells.

## Evaluation of cell-to-cell protein transport using the microfluidic device

By taking advantage of the microfluidic device in fixing cell position and monitoring cell behaviors, we next examine cell-to-cell protein transport in BY-2 cells. We performed FRAP experiments using fluorescent proteins to evaluate PD permeability. The cytoplasm of each cell in a single filament is separated by cell wall and molecules in the cytoplasm move between the cells via the PDs. In this study, BY-2 cells expressing green fluorescent protein (GFP) (ca 27 kDa) was firstly used to test PD permeability. The size exclusion limit of PDs in plant tissues is varied. Cell-to-cell movement of proteins with a molecular mass up to 50 kDa in sink tissues such as young leaves, root tips and ovules has been found [29, 30]. We therefore expected that cell-to-cell movement of GFP proteins would be detected.

BY-2 cells expressing GFP were sorted and introduced into the microfluidic device according to the above procedure. After an overnight curing culture, the medium in the microfluidic device was replaced with medium supplemented with cycloheximide at a final concentration of 50 μM to prevent new protein synthesis. Then, some or all of the cells of filaments were irradiated with 488 nm argon laser at the maximum power (25.0 mW) using a confocal laser microscope (LSM) to bleach the fluorescence of GFP (Fig 5A and 5B). We firstly confirmed

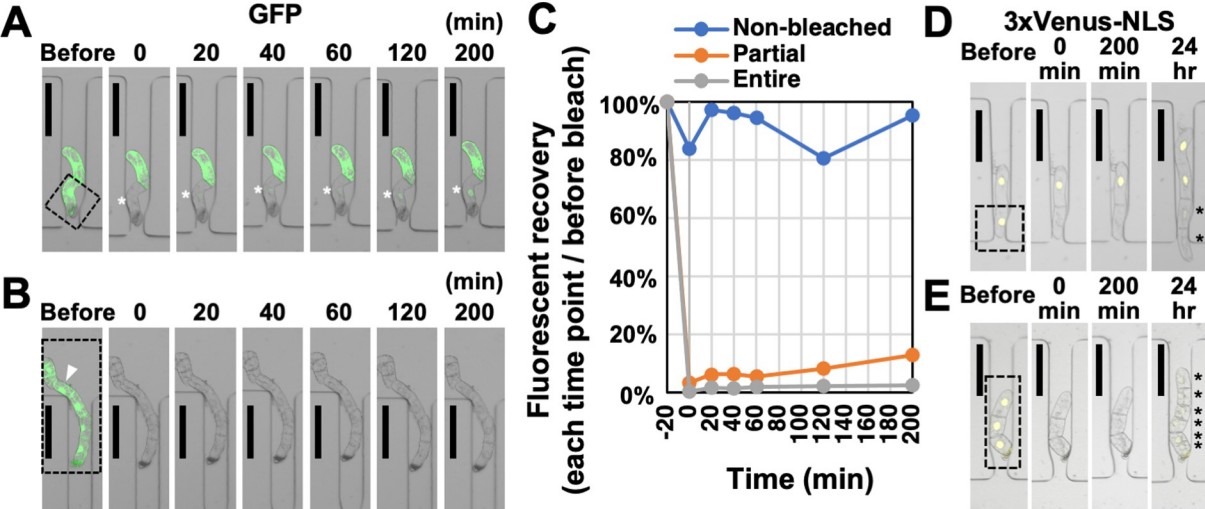

**Fig 5. Evaluation of plasmodesmata permeability by FRAP.** (A,B) Representative images of fluorescence recovery after photobleaching (FRAP) of BY-2 cells expressing green fluorescent protein (GFP) with cycloheximide. (A) Fluorescence recovery of filament, which only bleached a part. (B) Fluorescence recovery of filament that bleached the whole. Arrowhead indicates the cell analyzed in (C). (C) Measurement of fluorescence intensity by FRAP. The percentage of fluorescence intensity in each time point per that of before photobleaching was plotted. Blue (Non-bleached), the non-bleached cell in (A); orange (Partial), the bleached cell in (A); gray (Entire), the bleached cell in (B). (D,E) Representative images of FRAP of BY-2 cells expressing 3xVenus-NLS without cycloheximide. (D) Fluorescence recovery of filament, which only bleached a part. (E) Fluorescence recovery of filament that bleached the whole. Dashed rectangles indicate laser-irradiated areas. LSM images were merged from 8 consecutive optical sections. Fluorescence images were merged with bright-field images. Asterisks represent the recovery of fluorescence. Scale bars, 100 μm.

that fluorescence was almost invisible by 20 min laser irradiation (0 min). Next, since it has been reported that the recovery of free GFP without subcellular localization after FRAP can be confirmed from about 20 min [31], we looked for the recovery of fluorescence at 505 nm from 20 min to 200 min. Photobleaching was performed by irradiation of the entire cell area just inside the cell walls to avoid a negative effect on the permeability of PDs. Because the fluorescence intensity of intracellular region is not homogeneous, we measured the fluorescence intensity of the region surrounding the outline of the nucleus only. In the case that a cell filament was partially photobleached, a slight recovery of fluorescence in the bleached cells was observed after 20 min, and the fluorescence intensity increased over time (Fig 5A and 5C). On the other hand, fluorescence recovery was not observed in the cells from the cell filament which was entirely photobleached (Fig 5B and 5C). These observations suggest that the fluorescence recovery in the bleached cells was due to influx of free GFP from the neighboring cells through the PDs, not to novel GFP protein synthesis. Next, we evaluated the cell lines expressing 3×Venus-NLS proteins, whose molecular size (ca 82 kDa) is more than exclusion limit of PDs. In contrast to free GFP, in BY-2 cells expressing 3×Venus-NLS, no fluorescence recovery was observed by 200 min not only when all cells in the filament were bleached, but also when part of the cells were bleached. In both cases, since cycloheximide was not added, fluorescence of 3×Venus-NLS was restored in the bleached cells and their dividing daughter cells at 24 hours after photobleaching (Fig 5D and 5E). This result suggests that the large sized and nuclear localized 3×Venus-NLS proteins fail to pass the PDs.

## Conclusion

We have developed a microfluidic device to trap short filaments of BY-2 cells. The advantage of this method is traceability of multiple cell filaments for several days undergoing cell division and elongation. Using trapped BY-2 cells expressing fluorescent proteins and a FRAP technique, we evaluated the permeability of PDs. This technology will be useful to test the effect of various compounds on cell viability, cell division activity and cellular plastic properties, such as PD permeability for molecular transports. Not only FRAP experiment, photoconvertible fluorescent proteins will gain outcomes by shortening the laser irradiation time and increasing number of observations. Combining this method with genetically modified cell lines and/or super-resolution imaging technologies with marker proteins or biosensors will allow for further in-depth studies on the molecular mechanisms underlying the function and regulation of various cellular events. By taking advantage of this technique where the cells are oriented and immobilized in microchannels, effect of chemical agent on a broad cell biology can be finely traced in future study.

## Supporting information

**S1 Fig. Procedure for image analysis of BY-2 cells using ImageJ.**
(TIFF)

**S2 Fig. The distribution of the lengths of the minor and major axes of the approximate minimum ellipse of the cell filaments.**
(TIFF)

**S3 Fig. Cell division observed during culture in microfluidic devices.** BY-2 cells expressing 3xVenus-NLS were observed. Arrowheads indicate the nuclei in anaphase or telophase. LSM images were merged from 8 consecutive optical sections. Fluorescence images were merged with bright-field images. Scale bars, 50 μm.
(TIFF)

**S1 Dataset. Data for Fig 1C.**
(XLSX)

**S2 Dataset. Data for Fig 3A and S2 Fig.**
(XLSX)

**S3 Dataset. Data for Fig 3C.**
(XLSX)

**S4 Dataset. Data for Fig 4C.**
(XLSX)

**S5 Dataset. Data for Fig 4D.**
(XLSX)

## Acknowledgments

We thank Kazuki Yakamoto, Haruo Kassai, Ikue Yoshikawa, and Mayumi Taniguchi for technical assistance.

## Author Contributions

**Conceptualization:** Kazunori Shimizu, Hiroyuki Honda, Michitaka Notaguchi.

**Data curation:** Ken-ichi Kurotani.

**Formal analysis:** Yaichi Kawakatsu, Ken-ichi Kurotani, Masahiro Kikkawa, Ryo Tabata.

**Funding acquisition:** Ken-ichi Kurotani, Ryo Tabata, Michitaka Notaguchi.

**Investigation:** Kazunori Shimizu, Yaichi Kawakatsu, Ken-ichi Kurotani, Masahiro Kikkawa, Ryo Tabata, Michitaka Notaguchi.

**Methodology:** Yaichi Kawakatsu, Masahiro Kikkawa.

**Project administration:** Ken-ichi Kurotani, Hiroyuki Honda.

**Resources:** Daisuke Kurihara.

**Supervision:** Kazunori Shimizu, Hiroyuki Honda, Michitaka Notaguchi.

**Validation:** Ken-ichi Kurotani.

**Visualization:** Ken-ichi Kurotani.

**Writing – original draft:** Kazunori Shimizu.

**Writing – review & editing:** Ken-ichi Kurotani, Michitaka Notaguchi.

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
