## [Decision Letter · Decision Letter 0]

23 Mar 2022

PONE-D-22-04583Development of Microfluidic Chip for Entrapping Tobacco BY-2 CellsPLOS ONE

Dear Dr. Notaguchi,

Thank you for submitting your manuscript to PLOS ONE. After careful consideration, we feel that it has merit but does not fully meet PLOS ONE’s publication criteria as it currently stands. Therefore, we invite you to submit a revised version of the manuscript that addresses the points raised during the review process.

We look forward to receiving your revised manuscript.

Kind regards,

Hiroshi Yabu, PhD

Academic Editor

PLOS ONE

Journal Requirements:

"This work was supported by grants from the Japan Society for the Promotion of Science Grants-in-Aid for Scientific Research (18K05373 and 20H05501 to RT, 18KT0040, 18H03950, and 21H00368 to MN and 20H03273 to KK and MN), Grant-in-Aid for Scientific Research on Innovative Areas (20H05358 to DK), the Japan Science and Technology Agency PRESTO program (JPMJPR18K4 to DK), and the Canon Foundation (R17-0070 to MN)." 

5. Thank you for stating the following in the Funding Section of your manuscript: 

"This work was supported by grants from the Japan Society for the Promotion of Science Grants-in-Aid for Scientific Research (18K05373 and 20H05501 to RT, 18KT0040, 18H03950, and 21H00368 to MN and 20H03273 to KK and MN), Grant-in-Aid for Scientific Research on Innovative Areas (20H05358 to DK), the Japan Science and Technology Agency PRESTO program (JPMJPR18K4 to DK), and the Canon Foundation (R17-0070 to MN)."

"This work was supported by grants from the Japan Society for the Promotion of Science Grants-in-Aid for Scientific Research (18K05373 and 20H05501 to RT, 18KT0040, 18H03950, and 21H00368 to MN and 20H03273 to KK and MN), Grant-in-Aid for Scientific Research on Innovative Areas (20H05358 to DK), the Japan Science and Technology Agency PRESTO program (JPMJPR18K4 to DK), and the Canon Foundation (R17-0070 to MN)."

Reviewers' comments:

Reviewer's Responses to Questions

**Comments to the Author**

1. Is the manuscript technically sound, and do the data support the conclusions?

Reviewer #1: Yes

Reviewer #2: Yes

2. Has the statistical analysis been performed appropriately and rigorously? 

Reviewer #1: Yes

Reviewer #2: Yes

3. Have the authors made all data underlying the findings in their manuscript fully available?

Reviewer #1: Yes

Reviewer #2: Yes

4. Is the manuscript presented in an intelligible fashion and written in standard English?

Reviewer #1: Yes

Reviewer #2: Yes

5. Review Comments to the Author

Reviewer #1: The authors prepared microfluidic device to trap BY-2 cells and to evaluate physiological activity. Basically, the work shown in the manuscript is technically sounds and give a new device to evaluate cells especially those having filament-like morphology. But some points listed below should be revised before publication.

1) The authors designed microfluidic channels as shown in Figure 2, but what is the principle of the design? In order to trap cells the channel sizes and shapes should be well-considered and it should has specific design. What is the key parameters to determine the design of microfluidic channels?

2) Basically, the end of channel is open, therefore most of cells can be flow away and did not entrapped in the channels. Why did the authors design this type of open channels?

3) Regarding photobleaching, the recovery of fluorescent was not obvious. Is it sufficient intensity to say "recover"? Furthermore, please explain how the authors detect those slight difference.

4) Figure resolution is not high enough. Please upload more clear images.

Reviewer #2: This paper describes a study of trapping plant cells with a microchannel device for in-situ culture and observation. The cell filament trapped in the microchannel were shown to retain cell proliferation ability in it. Furthermore, the authors have succeeded in obtaining important information on cell-to-cell protein transport from fluorescence observations such as FRAP.

Therefore, this manuscript should be published after some minor revisions,

(1) Page12, Line207, the authors claim that the entrapment frequency is 22.3%. On the other hand, in Ref.28, although it is a microparticle, it is trapped more efficiently. It is necessary to describe the difference from the case of cell filaments.

(2) In fig.3C More than half of the trapped filaments consists of two cells. Why is the number 2?, Is there a correlation with the length of the minor axis of the approximate minimum ellipse?

(3) In fig.4D the mitotic indexes of the trapped BY-2 cell is slightly larger than that of the sorted filament. Why?

(4) In the FRAP experiment, please describe the wavelength and power of the irradiation laser light.

(5) Line 285, the remained rate of the filament during cultivation in microdevice is discussed. The authors claim that it is likely due to medium flow. However, the number of filament cells after 4 days increased 13.6 times (Fig. 4C), so is the effect possible?

(6) In Fig. 5A, is the photobleached area part of a “single cell”?

6. PLOS authors have the option to publish the peer review history of their article (what does this mean?). If published, this will include your full peer review and any attached files.

Reviewer #1: No

Reviewer #2: **Yes: **Yasutaka Matsuo

---

## [Author Response · Author response to Decision Letter 0]

29 Mar 2022

For Reviewer #1’s comments,

1) The authors designed microfluidic channels as shown in Figure 2, but what is the principle of the design? In order to trap cells the channel sizes and shapes should be well-considered and it should has specific design. What is the key parameters to determine the design of microfluidic channels?

2) Basically, the end of channel is open, therefore most of cells can be flow away and did not entrapped in the channels. Why did the authors design this type of open channels?

Response:

We appreciate the reviewer’s careful reading and providing thoughtful comments to our study. For the design of the microfluidic channels, the determination of parameters and the layout of the channels were based on the design of a previous study, Kim et al (Lab Chip, 2014). The key factor in this device design was the ratio of the width of each channel. According to their fluidic simulation, a trap zone was placed between the main channel and the side channel, and the pressure difference between the two attracted cells to the trap zone. We also modified the width according to the size of the cells. The size parameters of the microfluidic channel were based on their result that the ratio of the width of the inlet and outlet of trap zone, main channel, and side channel were critical for cell trapping.

3) Regarding photobleaching, the recovery of fluorescent was not obvious. Is it sufficient intensity to say "recover"? Furthermore, please explain how the authors detect those slight difference.

Response:

We have replaced the paper cited in the FRAP experiment with a more appropriate one. We consider the values in previous report, Martens et al. (Plant Physiol., 2010) on fluorescence recovery using FRAP to be in a similar range. (When photobleaching reduced fluorescence intensity in intercellular FRAP to about 15% of that before bleaching, it recovered to about 25% after 20 minutes.) The fluorescence values were quantified in ImageJ from images taken with confocal laser microscopy with the same observation parameters.

4) Figure resolution is not high enough. Please upload more clear images.

Response:

Image resolution was obtained at the optimal value for each observation in our experimental systems. The blurred focus was due to the thickness of the PDMS resin.

For Reviewer #2’s comments,

(1) Page12, Line207, the authors claim that the entrapment frequency is 22.3%. On the other hand, in Ref.28, although it is a microparticle, it is trapped more efficiently. It is necessary to describe the difference from the case of cell filaments.

Response:

We appreciate the reviewer’s careful reading and providing some suggestions. We added sentences “The width of the trap zone was determined from the standard short axis size of BY2 cells in the 1-cell state. The efficiency to trap BY-2 cell filaments in the trap zones was smaller than previously reported for Giardia cysts, probably because of the geometry complexity of filament-forming BY-2 cells [28]”

(2) In fig.3C More than half of the trapped filaments consists of two cells. Why is the number 2?, Is there a correlation with the length of the minor axis of the approximate minimum ellipse?

Response:

We apologize for the missing figure citation in Fig 3A, which was difficult to understand, and correct it. And the distribution of the lengths of the minor and major axes of the approximate minimum ellipse of the cell filaments extracted in Fig 3A was added as S2 Fig. As you said, there is some correlation between the length of the short axis and the number of cells in the filament. As shown in Fig. 3A and new S2 Fig, by passing the cell strainer through the cell strainer, we succeeded in collecting cell filaments with approximate minimum ellipse lengths of less than 60 µm and normally distributed with a peak at 35 µm. Filaments composed of two cells may explain much of this distribution. 

(3) In fig.4D the mitotic indexes of the trapped BY-2 cell is slightly larger than that of the sorted filament. Why?

Response:

This difference may not be significant, although the possibility cannot be ruled out that the supply of nutrients, oxygen, etc. was better than under normal culture conditions.

(4) In the FRAP experiment, please describe the wavelength and power of the irradiation laser light.

Response:

Thank you for pointing this out. More details are added in Materials and methods and main text.

(5) Line 285, the remained rate of the filament during cultivation in microdevice is discussed. The authors claim that it is likely due to medium flow. However, the number of filament cells after 4 days increased 13.6 times (Fig. 4C), so is the effect possible?

Response:

There was no large medium flow when fluorescence observation was being performed but it was possible during daily exchange of the liquid medium by pipetting. We haven’t observed the events during pipetting, it was not clear how the trapped filaments were actually lost. Therefore, we added sentences “Although it was not determined how the trapped cell filaments were actually lost, a large and fast flow of liquid medium during pipetting may cause removal of the trapped BY-2 cell filaments from the entrapped zones.”

(6) In Fig. 5A, is the photobleached area part of a “single cell”?

Response:

The area enclosed by a square targeting almost one entire cell was photobleached, except for the boundary with the adjacent cell.

---

## [Editor Report · Decision Letter 1]

31 Mar 2022

Development of Microfluidic Chip for Entrapping Tobacco BY-2 Cells

PONE-D-22-04583R1

Dear Dr. Notaguchi,

We’re pleased to inform you that your manuscript has been judged scientifically suitable for publication and will be formally accepted for publication once it meets all outstanding technical requirements.

Kind regards,

Hiroshi Yabu, PhD

Academic Editor

PLOS ONE

Additional Editor Comments (optional):

The authors responded to whole queries from reviewers and I recommend to accept the manuscript to be accepted in PLoS One.

---

## [Editor Report · Acceptance letter]

5 Apr 2022

PONE-D-22-04583R1 

Development of Microfluidic Chip for Entrapping Tobacco BY-2 Cells 

Dear Dr. Notaguchi:

I'm pleased to inform you that your manuscript has been deemed suitable for publication in PLOS ONE. Congratulations! Your manuscript is now with our production department. 

Kind regards, 

on behalf of

Dr. Hiroshi Yabu 

Academic Editor

PLOS ONE